# SWAT-NN: Simultaneous Weights and Architecture Training for Neural Networks in a Latent Space

**Zitong Huang**
Electrical and Computer Engineering
University of Southern California
Los Angeles, CA 90007
chuang95@usc.edu

**Mansooreh Montazerin**
Electrical and Computer Engineering
University of Southern California
Los Angeles, CA 90007
mmontaze@usc.edu

**Ajitesh Srivastava**
Electrical and Computer Engineering
University of Southern California
Los Angeles, CA 90007
ajiteshs@usc.edu

## Abstract

Designing neural networks typically relies on manual trial and error or a neural architecture search (NAS) followed by weight training. The former is time-consuming and labor-intensive, while the latter often discretizes architecture search and weight optimization. In this paper, we propose a fundamentally different approach that simultaneously optimizes both the architecture and the weights of a neural network. Our framework first trains a universal multi-scale autoencoder that embeds both architectural and parametric information into a continuous latent space, where functionally similar neural networks are mapped closer together. Given a dataset, we then randomly initialize a point in the embedding space and update it via gradient descent to obtain the optimal neural network, jointly optimizing its structure and weights. The optimization process incorporates sparsity and compactness penalties to promote efficient models. Experiments on regression datasets demonstrate that our method effectively discovers sparse and compact neural networks with strong performance.

## 1 Introduction

Traditionally, designing neural networks involves a time-consuming, trial-and-error process: fixing an architecture, training it, evaluating performance, and iteratively refining it based on expert intuition. To address this challenge, Neural Architecture Search (NAS) has emerged as a promising framework for automatically discovering effective neural network architectures Ren et al. [2021], Elsken et al. [2019]. Existing NAS methods can be broadly classified into discrete and continuous search approaches. Discrete search category predominantly employed Reinforcement Learning(RL) Zoph and Le [2016] and evolutionary algorithms Liu et al. [2021], Stanley and Miikkulainen [2002]. Both approaches operate over high-dimensional discrete representations and navigate a vast combinatorial search space defined by a predefined set of operations. In contrast, continuous NAS methods Elsken et al. [2019], Liu et al. [2018] have gained attention for their efficiency and ability to operate in a differentiable search spaceSantra et al. [2021], Xie et al. [2018], Cai et al. [2018]. Nevertheless, while continuous NAS methods reduce search cost and enable differentiable optimization, they are often constrained by task-specific predictors or datasets, and typically search for architectures and weights in a decoupled or alternating manner.

In this work, we propose SWAT-NN, which jointly optimizes both the architecture and weights to achieve high performance on given datasets. It is based on a multi-scale autoencoder framework, trained to embed functionally similar neural networks close to one another within a continuous latent space. The encoder learns to represent the full neural network – including both its structure and parameters – in an embedding space, from which multiple decoders generate functionally similar neural networks of varying depths. This embedding, once trained, enables gradient-based optimization by optimizing task-specific performance. Because the latent representation jointly encodes architecture and weights, our method performs unified optimization over both. We evaluate our proposed approach in the context of multi-layer perceptrons (MLPs) with three different activation functions: sigmoid, tanh, and leaky ReLU, and leaky ReLU, and apply it to the Continuous Optimization Benchmark Suite from Neural Network Regression (CORNN) benchmark, which contains 54 regression datasets derived from diverse benchmark function datasets Malan and Cleghorn [2022]. Notably, the proposed approach diverges from conventional NAS paradigms by treating networks holistically as function approximators, rather than representing them as graphs composed of discrete operations. This allows the embedding to be over their actual functional behavior. Our contributions are:

1. We propose a new framework, SWAT-NN, that performs simultaneous optimization over neural network architectures and weights within a universal embedding space, enabling fine-grained design choices such as neuron-level activation and adaptive layer widths – unlike traditional NAS methods that decouple architecture search and weight tuning.

2. Through extensive experiments on 54 regression tasks, we show that SWAT-NN discovers significantly sparser and more compact models compared to existing methods, while maintaining comparable or better accuracy.

## 2 Related Work

Traditional NAS approaches include reinforcement learning (RL) Zoph and Le [2016] or evolutionary algorithms Liu et al. [2021], Stanley and Miikkulainen [2002]: RL formulates architecture design as a sequential decision-making process, whereas evolutionary methods construct architectures by iteratively mutating or recombining predefined operations. However, these methods operate in a discrete search space, making them computationally expensive and hard to scale. More recent efforts explore continuous NAS, which relaxes the discrete architecture space into a continuous domain.

Differentiable Architecture Search (DARTS) Liu et al. [2018] relaxes discrete architectural choices by introducing a softmax over all candidate operations on each edge of a directed acyclic graph (DAG). During the search process, the network weights are optimized on the training set, while the architecture parameters are updated on the validation set. Such decoupling breaks joint optimization and can result in suboptimal architectures due to misaligned objectives between weight and architecture updates.

Neural Architecture Optimization (NAO) based on a graph variational autoencoder (VAE) Li et al. [2020] encodes network architectures into a continuous latent representation specific to the given set of datasets. It uses a surrogate performance predictor, a regression model estimating task-specific accuracy of an architecture based on its latent representation, to guide the search. However, NAO fully decouples architecture discovery from weight optimization and relies on dataset-specific predictors. Further, it requires retraining of the autoencoder from scratch for each new set of tasks.

In contrast, we propose a novel approach that departs from conventional NAS formulations. Rather than searching over discrete or graph-based compositions of operations, we treat entire neural networks as function approximators and embed their complete mathematical behavior – including both architecture and weights – into a universal continuous latent space. Optimization is performed directly on this functional representation. Moreover, our method is agnostic to specific datasets, enabling more general and flexible network discovery.

## 3 Methodology

The key idea of SWAT-NN is based on achieving the following two goals: (1) training a multi-scale autoencoder to construct a latent embedding space for MLPs (subsection 3.1); and (2) searching for sparse and compact networks via gradient descent in the learned embedding space (subsection 3.2).

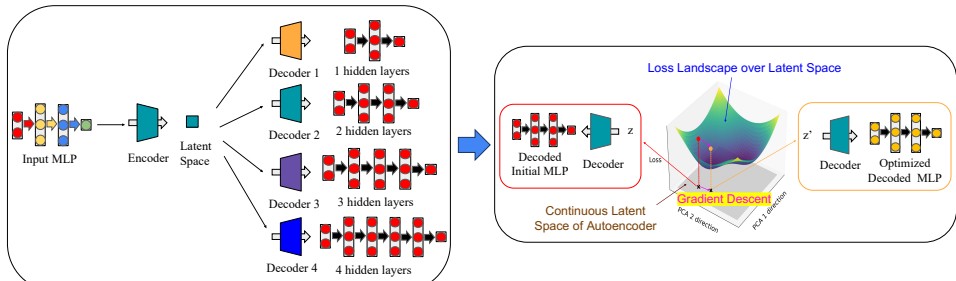

Figure 1: Left: Multi-scale autoencoder architecture with four decoders, each corresponding to MLPs with 1–4 hidden layers. Right: Pipeline for training optimal MLPs through gradient-based search in the continuous latent space learned by the autoencoder.

## 3.1 Embedding MLPs with Autoencoder

We aim to encode both the architecture and corresponding weights of a neural network into a shared embedding space through a multi-scale autoencoder framework. The encoder, denoted as $E$, maps a given MLP into a latent representation. A set of $L$ decoders, denoted as $D_1, D_2, \ldots, D_L$, then, generate MLPs with 1 to $L$ hidden layers, respectively. The autoencoder is trained to generate MLPs that may not be reconstructions of the input networks, but are functionally similar. To enable autoencoder training, we first transform each neural network's architecture and weights into a structured, fixed-size matrix representation. This representation serves as the input to the encoder. For details on the matrix encoding process, see Appendix B in the Supplementary Material.

**Multi-scale Autoencoder.** The multi-scale autoencoder consists of a single encoder $E$ and decoders $D_1, D_2, \ldots, D_L$, where each decoder $D_k$ generates an MLP with $k$ hidden layers. The overall structure is illustrated in Figure 1.

The encoder and decoders are implemented using a GPT-2 architecture Radford et al. [2019]. Following the matrix representation of MLPs described in Appendix B in the Supplementary Material, each row $h$ corresponds to the concatenation of all outgoing weights from the $h^{\text{th}}$ neuron across every layer of the MLP. Each row $h$ is, then, linearly projected to be compatible with the token embedding size of the GPT-2. During the encoding stage, the MLP matrix representation is provided as input to the encoder, which maps it to a latent embedding. In the decoding stage, each decoder produce a new matrix representation of an MLP. For decoder $D_k$, we retain only the first $k$ hidden-layer blocks from the generated output matrix, representing of an MLP with $k$ hidden layers. Detailed training objectives and complexity analysis of the autoencoder are discussed in Appendix C in the Supplementary Material.

## 3.2 Training MLP in the Embedding Space

Given a dataset $S$, we leverage the autoencoder to search for a sparse and compact MLP tailored to $S$. The decoded MLP can vary in the number of hidden layers (up to $L$), the number of neurons per layer (up to $N$), sparsity patterns, and activation functions across neurons (selected from $\mathcal{O}$).

**Training Optimal MLPs for a Dataset.** The learnable parameters of the multi-scale autoencoder are fixed during this stage. Since all MLPs are embedded into a continuous latent space, the problem of finding an optimal MLP reduces to optimizing an embedding vector $z \in \mathbb{R}^d$ within this space. Given a dataset $S = \{(x, y)\}$, we define the objective function for decoder $D_i$ as:

$$L_i(z) = \sum_{(x,y) \in S} \left( [D_i(z)](x) - y \right)^2 \tag{1}$$

where $[D_i(z)](x)$ denotes the output of the decoded MLP with $i$ hidden layers evaluated at input data $x$. To solve this optimization problem, we initialize $z$ by randomly sampling a point in the embedding space and performing gradient descent to minimize $L_i(z)$ with respect to $z$. $\mathcal{O}(L \cdot T \cdot f)$, where $L$ is the number of decoders, $T$ is the number of gradient steps per decoder, and $f$ is the cost of a forward or backward pass through a GPT-2 decoder. Details on sparsity regularization and activation selection are provided in Appendix D in the Supplementary Material.

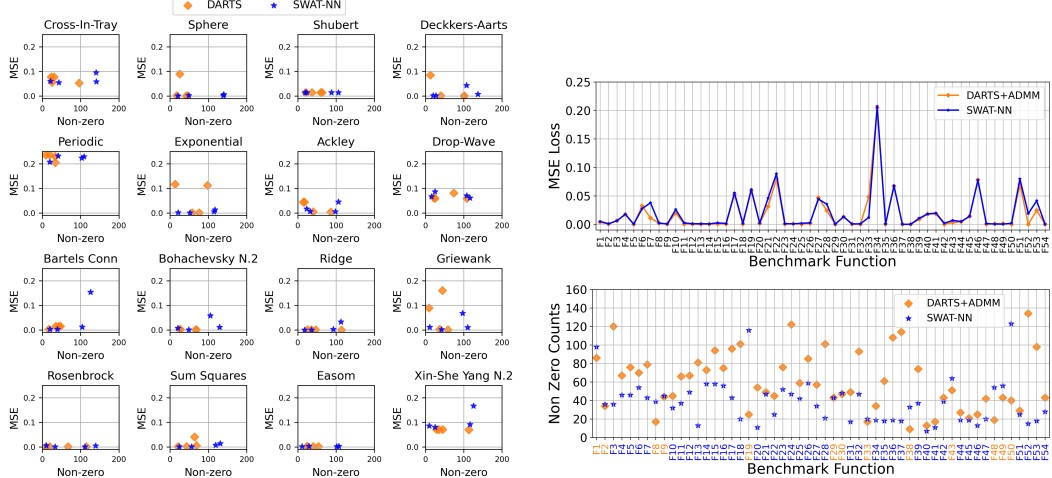

Figure 2: Left: MSE vs non-zero weights between DARTS+ADMM and SWAT-NN across 16 randomly selected benchmark function datasets. Each point represents a model configuration. Right: Performance of best models on all 54 datasets, showing Test MSE (top) and corresponding non-zero weights (bottom). Color coding highlights which method yields the more compact architecture.

## 4 Experiments and Results

Our main experiment evaluates whether SWAT-NN can jointly optimize activation functions, neuron usage, and weight sparsity to produce sparse and compact MLPs. Experiments are conducted on 54 regression datasets from the CORNN benchmark suite Malan and Cleghorn [2022]. All the code is publicly available [1].

**Baseline.** Although no prior work jointly optimizes both architecture and weights, we compare against a baseline, evaluated in subsection 4.1, that adapts DARTS to MLPs first to search over number of neurons per layer and neuron-level activation functions, and then prunes using Alternating Direction Method of Multipliers (ADMM) method. We train the DARTS model for 50 epochs, then applied ADMM with $\rho = 2$ and a threshold of $10^{-1}$ to sparsify the weights. These hyperparameters are selected after increasing sparsity strength until performance on roughly one-third of the tasks dropped over 5% – our tolerable limit.

**Autoencoder Settings.** We trained a multi-scale GPT-2 autoencoder with one encoder and four decoders for MLPs with 1-4 hidden layers. Each input MLP has two inputs, one output, and up to 7 neurons per hidden layer. Weights and biases are sampled from $[-5, 5]$ and $[-1, 1]$, respectively. For each MLP, 1000 inputs are uniformly sampled from $[-1, 1]^2$, and the autoencoder is trained using the loss function in Appendix C of the Supplementary Material. Each training epoch includes 50,000 batches (batch size 64) and takes around 3 hours to complete on a single RTX A5000 GPU. In Appendix F.5 of the Supplementary Material, we also incorporate varying input and output dimensions by applying the same masking and padding strategy to the boundary layers.

### 4.1 Learning Sparse and Compact MLPs

In SWAT-NN setting, for sparsity penalty, we set $\lambda_s$ to $1 \times 10^{-4}$, $\mu_1 = 0.1$ for the $\ell_1$ penalty and $\mu_c = 0.01$ for the soft count term. For neuron compactness penalty, we use $\alpha = 0.4$ and $\beta = 0.001$. For activation function selection, we set $T_{\text{init}} = 1.0$, $T_{\text{final}} = 0.01$, $E_{\text{anneal}} = 3000$. These hyperparameters are tuned using the last two benchmark functions (F53 and F54) from the CORNN dataset as a validation set. SWAT-NN training takes about 1 minute per task and per MLP configuration on a machine with a single RTX A5000 GPU, 128GB RAM, and a 56-core CPU.

---

[1] https://github.com/zitonghuangcynthia/SWAT-NN.git

The left plot in Figure 2 presents MSE vs non-zero weights for 16 benchmark function datasets, where each point represents a model configuration, and positions closer to the bottom-left indicate lower MSE and higher sparsity. SWAT-NN always identifies solutions closer to the optimal accuracy-sparsity trade-off, demonstrating its advantage in producing more compact and efficient networks.

The right plot in Figure 2 show the results on all dataset in terms of MSE and number of non-zero weights. For each function, we apply a two-step selection criterion: we first identify architectures whose MSE is within 5% of the minimum, and among them select the one with the fewest non-zero weights. While SWAT-NN and DARTS followed by ADMM achieve similar test accuracy, in most cases, SWAT-NN consistently discovers models with significantly fewer non-zero weights.

## 5 Conclusion

We presented SWAT-NN, a novel framework that simultaneously optimizes both neural architectures and their weights within a universal and continuous embedding space. Through experiments on 54 regression datasets, we demonstrated that SWAT-NN achieves comparable or better accuracy while producing more compact and sparse neural networks than state-of-the-art methods. While this work focuses on MLPs, the SWAT-NN framework can be extended to more complex neural network families such as Temporal Convolutional Networks (TCNs), Convolutional Neural Networks (CNNs), and Recurrent Neural Networks (RNNs).

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
