# Supplementary Material for "SWAT-NN: Simultaneous Weights and Architecture Training for Neural Networks in a Latent Space"

**Zitong Huang**
Electrical and Computer Engineering
University of Southern California
Los Angeles, CA 90007
chuang95@usc.edu

**Mansooreh Montazerin**
Electrical and Computer Engineering
University of Southern California
Los Angeles, CA 90007
mmontaze@usc.edu

**Ajitesh Srivastava**
Electrical and Computer Engineering
University of Southern California
Los Angeles, CA 90007
ajiteshs@usc.edu

## A Overall Framework of Autoencoder and Gradient Descent

Figure 1 illustrates the overall pipeline of SWAT-NN, including the multi-scale autoencoder design and the downstream gradient-based search procedure used to generate optimal MLPs.

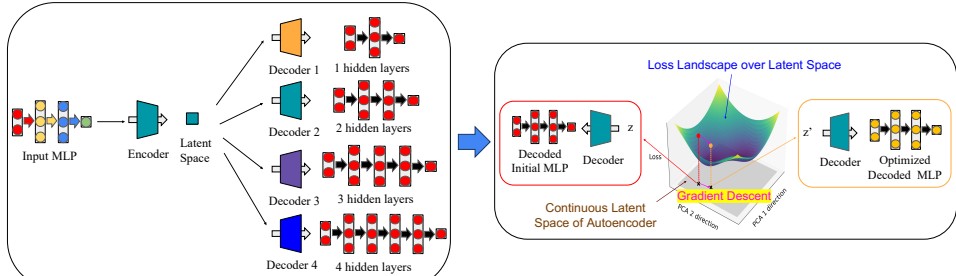

Figure 1: Left: Multi-scale autoencoder architecture with four decoders, each corresponding to MLPs with 1–4 hidden layers. Right: Pipeline for training optimal MLPs through gradient-based search in the continuous latent space learned by the autoencoder.

## B More Detailed Matrix Representation of Neural Networks

### B.1 Matrix Representation of MLPs

To train an autoencoder for encoding neural networks, we require a structured and fixed-size representation of MLPs that can act as input. Both the architecture and weights of an MLP are encoded in a concise matrix-based representation. For an MLP with $L$ hidden layers, each containing $N$ neurons, input dimension $i$, and output dimension $o$, the weights between consecutive layers can be expressed as a sequence of matrices:

$$[(W_{i,N}, b_N), (W_{N,N}, b_N), \ldots, (W_{N,N}, b_N), (W_{N,o}, b_o)] \tag{1}$$

Preprint.

where $W_{i,N} \in \mathbb{R}^{i \times N}, W_{N,N} \in \mathbb{R}^{N \times N}$, and $W_{N,o} \in \mathbb{R}^{N \times o}$ are weight matrices between layers, and the corresponding bias vectors are $b_N \in \mathbb{R}^{N \times 1}, b_o \in \mathbb{R}^{o \times 1}$. To ensure uniform dimensionality for concatenation, we apply zero-padding to the first and last matrices so that all matrices have dimensions of $N \times N$. In addition, each bias vector is padded to $N \times 1$ and horizontally concatenated into the matrix of weights. To differentiate between the actual weights and padded entries, a secondary mask matrix is appended in parallel to the matrix described above. More specifically, each position corresponding to an original MLP weight is assigned a value of 1 in the mask, while positions introduced through zero-padding are assigned a value of 0. The sizes of the input and output layers can also be flexibly specified using boundary-layer masks (see subsection F.5). This basic representation is later extended to incorporate varying numbers of neurons and activation functions in subsection B.2 and subsection B.3; an overview of the complete representation scheme is illustrated in Figure 2.

## B.2   Varying Number of Neurons

To enable the autoencoder to account for different numbers of neurons per layer as part of the network architecture, we introduce a neuron indicator matrix to explicitly encode this structural variation. For an MLP with $L$ hidden layers, each with at most $N$ neurons, we augment the matrix representation introduced in Section 3.1 of the main text by appending an additional mask matrix $M \in \{0,1\}^{N \times L}$. Each column of $M$ corresponds to one hidden layer, which indicates the activation status of all the neurons in that layer. Specifically, $M_{h,j} = 1$ if the $h$th neuron in the $j$th hidden layer is active in the input MLP, and $M_{h,j} = 0$ otherwise.

We apply the autoencoder introduced in Section 3.1 of the main text to this extended representation. The decoded neuron indicator matrix $\hat{M} \in [0,1]^{N \times L}$ is obtained via a sigmoid activation, where each element $\hat{M}_{h,j}$ represents the probability of the $h$th neuron in the $j$th hidden layer being active. To preserve the differentiability of the loss function during training, we avoid hard thresholding for enforcing binary activations. Instead, we use a soft thresholding method to decide entirely active vs pruned neurons described in subsection D.1.

## B.3   Changing Activation Functions

To capture activation function choices as part of the network architecture, we extend the matrix representation to explicitly encode which activation function is applied to each neuron. We incorporate activation function choices into the matrix representation introduced in subsection B.1 and subsection B.2 through appending additional columns that specify the activation function used by each neuron. Assuming a total of $A$ possible activation functions, the activation type of each neuron is encoded as a one-hot vector of length $A$. The final representation is:

$$
\begin{aligned}
[(W_{i,N}, b_N),\ &F_{N,A},\ (W_{N,N}, b_N),\ F_{N,A}, \\
&\ldots,\ (W_{N,N}, b_N),\ F_{N,A},\ (W_{N,o}, b_o),\ M_{N,L}]
\end{aligned}
\tag{2}
$$

The final matrix $M_{N,L} \in \{0,1\}^{N \times L}$ indicates neuron activity as described in subsection B.2. Each $F_{N,A} \in \{0,1\}^{N \times A}$ follows a hidden layer's weight and bias matrix and encodes the activation function for each neuron in that layer. An illustration of the whole matrix representation is shown in Figure 2.

After feeding the extended representation into the encoder-decoder framework, we apply continuous relaxation to decoded $\hat{F}_{N,A}$ matrices to enable differentiable selection of activation functions. Each row of $\hat{F}_{N,A}$ is interpreted as a softmax distribution over the $A$ candidate activation functions. Formally, for a neuron $h$ in layer $j$, the output is computed as a soft combination of candidate activation functions:

$$
y_{(h,j)}(x) = \sum_{k=1}^{A} \alpha_{(h,j),k} \cdot o_k(x)
\tag{3}
$$

where $x$ is the input flows through current neuron $(h,j)$, and $o_k(\cdot) \in \mathcal{O}$ denotes the $k$th candidate activation function from a predefined activation function set $\mathcal{O}$. The activations' selection probabilities are given by:

$$
\alpha_{(h,j),k} = \text{softmax}(\hat{F}_{h,:}^{(j)})_k
\tag{4}
$$

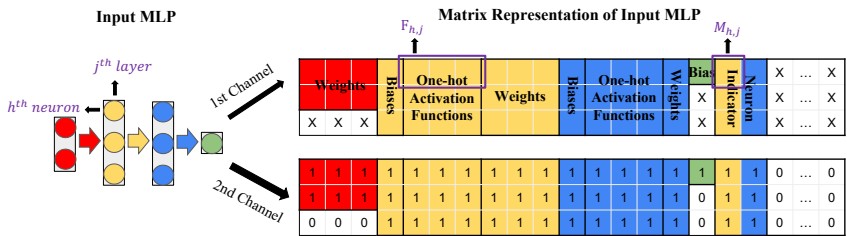

Figure 2: Matrix representation for a 2 hidden-layer MLP with 3 neurons per layer. Different colors indicate elements associated with neurons from different layers, encoding the outgoing weights, activation functions, and bias terms of each neuron. Entries marked with 'X' represent zero-padding.

Here, $\hat{F}_{h,:}^{(j)} \in \mathbb{R}^A$ denotes the decoded activation logits for the $h^{\text{th}}$ neuron in the $j^{\text{th}}$ hidden layer. This relaxation preserves differentiability during training, enabling smooth optimization over activation selections. A near-discrete activation assignment can be enforced at inference time in subsection D.2.

## C Details of Training the Autoencoder

We generate input vectors $x$, and feed them into the input MLP $\mathcal{N}_s$ to obtain the corresponding outputs. Each input MLP is then passed through the autoencoder to produce decoded MLPs with varying depths $\{1, 2, \ldots, L\}$. We adopt a minimum-loss objective to encourage functional similarity between the input MLP and its decoded variants. The loss function is defined as:

$$L_{ED} = \sum_{\mathcal{N}_s} \min_{i \in 1,2,\ldots,L} \left( \sum_x \left( \mathcal{N}_s(x) - [D_i(E(\mathcal{N}_s))](x) \right)^2 \right) \tag{5}$$

where $\mathcal{N}_s(x)$ denotes the output of the original input MLP evaluated at input $x$, and $[D_i(E(\mathcal{N}_s))](x)$ denotes the output of the decoded MLP with $i$ hidden layers, reconstructed from the bottleneck embedding and evaluated at the same input $x$. This objective encourages at least one decoded MLP to closely approximate the functional behavior of the original MLP across all sampled inputs. It effectively pulls functionally similar MLPs closer together in the embedding space.

The runtime complexity for training a single encoder and $L$ decoders is $\mathcal{O}\left(E \cdot \frac{N}{B} \cdot L \cdot f\right)$, where $f$ is the cost of a forward or backward pass through GPT-2, $N$ is the dataset size, $B$ is the batch size, and $E$ is the number of training epochs.

## D Details of Training MLPs in the Embedding Space

### D.1 Sparsity and Compactness

To promote neural networks that are both sparse (fewer connections) and compact (fewer active neurons), we incorporate two regularization terms into the loss: a weight sparsity penalty $\mathcal{P}_s$ to drive individual weights toward zero, and a neuron compactness penalty $\mathcal{P}_n$ to suppress redundant neurons and promote compact architectures. The overall loss function is defined as:

$$L_i(z) = \sum_{(x,y) \in S} \left( [D_i(z)](x) - y \right)^2 + \lambda_s \cdot \mathcal{P}_s + \mathcal{P}_n \tag{6}$$

where $\mathcal{P}_s$ denotes the sparsity penalty weighted by coefficients $\lambda_s$, and $\mathcal{P}_n$ denotes the neuron compactness penalty.

**Sparsity Penalty.** $\mathcal{P}_s$ consists of two components: an $\ell_1$-regularization term and a soft counting switch. Formally, it is defined as:

$$\mathcal{P}_s = \mathcal{L}_1 + \text{SoftCount}, \tag{7}$$

$$\text{Where, } \mathcal{L}_1 = \mu_1 \|W[D_i(z)]\|_1, \tag{8}$$

$$\text{And, SoftCount} = \mu_c \cdot \sigma\left(20 \cdot \|\|W[D_i(z)]\| - t_s\|_1\right) \tag{9}$$

Here, $W[D_i(z)]$ denotes the weight matrix of the decoded MLP $D_i(z)$, and $t_s$ is a learnable threshold parameter. The $\mathcal{L}_1$ term encourages weights to shrink toward zero. The soft counting switch approximates the number of weights below the threshold $t_s$, using a scaled sigmoid function to softly suppress small-magnitude weights. During training, weights smaller than $t_s$ are softly masked using a sigmoid-based function that smoothly pushes them toward zero; at test time, they are hard-masked. Both $\mu_1$ and $\mu_c$ are tunable hyperparameters that control the relative strength of $\ell_1$ regularization and the soft counting penalty, respectively.

**Neuron Compactness Penalty.** To reduce the number of active neurons per layer by pruning unnecessary ones, we define the neuron compactness penalty $\mathcal{P}_n$ as a combination of two terms: the negative variance and the average magnitude of soft activation masks across neurons within each layer:

$$\mathcal{P}_n = -\alpha \cdot \frac{1}{L} \sum_{i=1}^{L} \text{std}(M_i) + \beta \cdot \frac{1}{L} \sum_{i=1}^{L} \text{mean}(M_i) \tag{10}$$

Here, $M_i$ denotes the soft neuron activation indicators for the $i^{\text{th}}$ hidden layer, and $L$ is the number of layers. The first term encourages the activation probabilities within each layer to become more polarized, while the second term penalizes the average activation level to further suppress unnecessary neurons. Coefficients $\alpha$ and $\beta$ are hyperparameters controlling the strength of each term. During training, neuron indicators smaller than a fixed threshold $t_n = 0.5$ are softly masked using a sharpened sigmoid function that closely approximates hard thresholding while maintaining differentiability; at test time, neurons this threshold are hard-masked as inactive.

## D.2 Activation Function Selection

We adopt a temperature-controlled softmax mechanism Liu et al. [2018] to gradually fix the activation function assigned to each neuron. We define a temperature schedule over training epochs as follows:

$$T(e) = \max \left( T_{\text{final}}, T_{\text{init}} \cdot \left( 1 - \frac{e}{E_{\text{anneal}}} \right) \right) \tag{11}$$

Here, $T_{\text{init}}$ is the initial temperature at the beginning of training, $T_{\text{final}}$ is the minimum temperature reached. $E_{\text{anneal}}$ is the number of epochs over which the temperature linearly decays to $T_{\text{final}}$, and $e$ denotes the current training epoch.

During training, for each neuron, a softmax over activation function candidates (i.e., ReLU, Tanh, Sigmoid) is applied using the current temperature $T(e)$. Let $\mathbf{a}_{h,j} \in \mathbb{R}^A$ denote the unnormalized logits for $A$ activation candidates of the $h^{\text{th}}$ neuron in layer $j$; the contribution weight of activation operation $a$ at neuron $(h, j)$ is computed as:

$$\alpha_{\text{h,j,a}} = \frac{\exp\left(\mathbf{a}_{h,j,a}/T(e)\right)}{\sum\limits_{a=1}^{A} \exp\left(\mathbf{a}_{h,j,a}/T(e)\right)} \tag{12}$$

As training progresses and $T(e)$ decreases, the softmax distribution sharpens, eventually approximating a one-hot selection. At test time, we select the activation function with the highest softmax score for each neuron as its final activation.

## E Experiments on Activation Functions and Sparse MLPs

Further experiments evaluating the effectiveness of SWAT-NN are structured into two components:

- **Exp 1: Search for Activation Functions.** To isolate the effect of neuron-level activation function selection, we evaluate our framework described in subsection D.2 and compare it against two baselines: (a) autoencoder-based search with fixed activation functions, and (b) traditional MLP training with fixed activation functions.

- **Exp 2: Search for Sparse MLPs.** To further isolate the effect of weight sparsity, we fix neuron counts and optimize for sparse weights, as described in subsection D.1. We compare against a baseline where autoencoder-based search is followed by ADMM pruning.

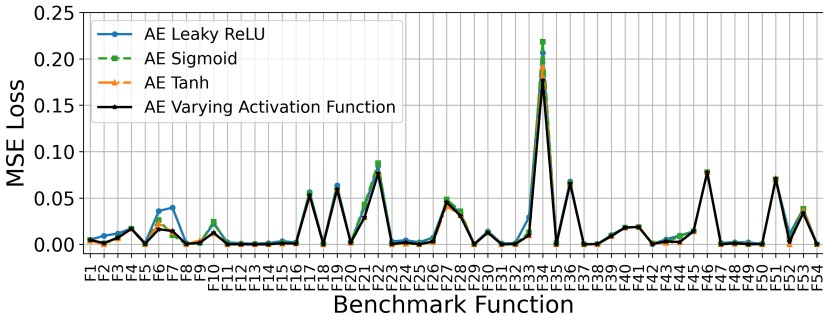

Figure 3: Comparison of MSE using autoencoder-based search with fixed activation functions versus neuron-level activation function search.

### E.1 Baselines

**Exp 2: Autoencoder-based Search with Fixed Activation Functions** Used as a baseline in subsection E.2, we fix the activation function for all neurons to one of Sigmoid, Tanh, or Leaky ReLU, and use the trained autoencoder to optimize over weights (8000 epochs, learning rate 0.1).

**Exp 2: Traditional MLP Training** Also evaluated in subsection E.2, we train MLPs with fixed activation functions and randomly initialized weights using standard gradient descent for 6000 epochs at a learning rate of 0.01. This reflects a common trial-and-error approach.

**Exp 3: Autoencoder with ADMM** Used as a baseline in subsection E.3, we combine autoencoder-based architecture and weight optimization without sparsity penalty with post-hoc ADMM pruning. Models are trained for 8000 epochs with a learning rate of 0.1, while the ADMM uses $\rho = 2$ and a pruning threshold of $10^{-1}$.

### E.2 Search for Optimal MLPs with Activation Functions

We focus exclusively on the search over activation functions. We use all 7 neurons in each hidden layer and do not apply the sparsity penalty discussed in subsection D.1 during training. The same hyperparameters for activation function selection are used as described in the Section 4.1 of the main text. Each experiment is repeated three times, and we report the average MSE across runs.

We compare our method, which searches neuron-level activation functions in the embedding space, with autoencoder-based search with fixed activation functions described in subsection E.1. As shown in Figure 3, our method consistently achieves performance comparable to or better than the best fixed-activation baselines, demonstrating that searching over activation types improves model performance.

Figure 4 further compares our method to the traditional training baseline described in subsection E.1. Our method achieves performance comparable to MLPs trained using traditional training method with fixed activation functions in most cases. While the accuracy is similar or slightly lower, the ability to flexibly assign activation functions per neuron enables our method to produce significantly more compact and sparse networks, as demonstrated in Section 4.1 of the main text and subsection E.3.

### E.3 Search for Sparse MLPs

Building upon the activation function search in subsection E.2, we compare our autoencoder-based search incorporated with sparsity penalties with the baseline of autoencoder-based search with ADMM afterwards described in subsection E.1. For our autoencoder-based method, we set $\lambda_s$ to $1 \times 10^{-3}$, and use $\mu_1 = 0.1$ for the $\ell_1$ penalty and $\mu_c = 0.01$ for the soft count term. Note that in this set of experiments, we do not apply neuron-level pruning yet.

The MSE vs non-zero weights scatter plots for 16 benchmark functions shown in Figure 5 indicates that our method consistently produces configurations that dominate or closely approach the left-down corner region. Meanwhile, Figure 6 summarize the best selected model configurations in terms of MSE and number of non-zero weights. Compared to the models without any pruning, our

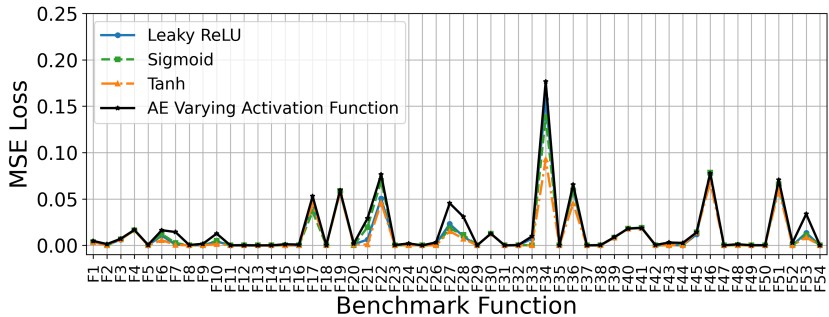

Figure 4: Comparison of the best MSEs obtained by traditional training baseline using Leaky ReLU, Sigmoid, and Tanh activations versus autoencoder-searched MLPs with neuron-level activation functions.

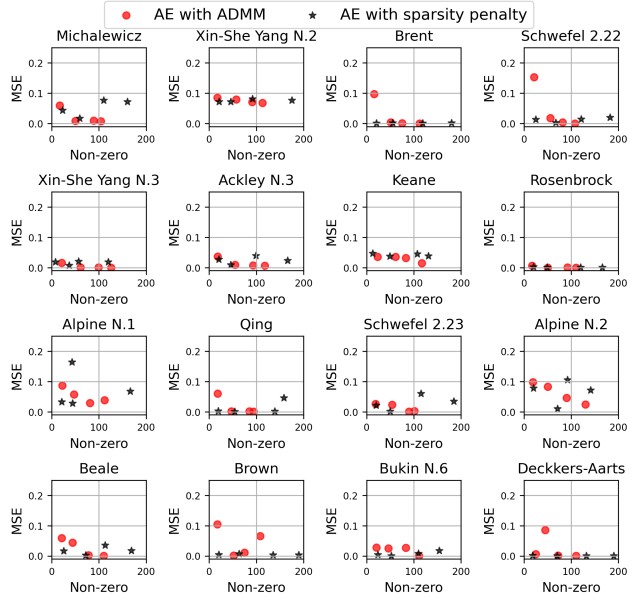

Figure 5: MSE vs. non-zero weights for autoencoder-based search with ADMM and with sparsity penalties. Each point represents a model configuration.

autoencoder-based method with sparsity penalties achieves substantial reductions in non-zero weights while maintaining similar MSE. Furthermore, relative to the baseline where autoencoder-based search is followed by ADMM pruning, our simultaneous search consistently identifies more compact models. In most cases, it also achieves lower or comparable test MSE, demonstrating the effectiveness of integrating sparsity directly into the optimization process rather than decomposing architecture discovery and weight training or pruning into separate stages.

## F Discussion

### F.1 Smoothness of the Embedding Space

The embedding space learned by the multi-scale autoencoder is central to our framework, mapping functionally similar MLPs to nearby points in the embedding space. A smooth embedding space enables accurate gradient-based navigation toward optimal networks. To assess smoothness of the embedding space for each decoder $D_k$, we sample a base embedding vector $z_{\text{init}} \sim \mathcal{N}(0, I)$ and generate 200 nearby embeddings by adding small Gaussian noise. We then apply principal component analysis (PCA) and extract the top two principal directions, denoted $v_1$ and $v_2$. These

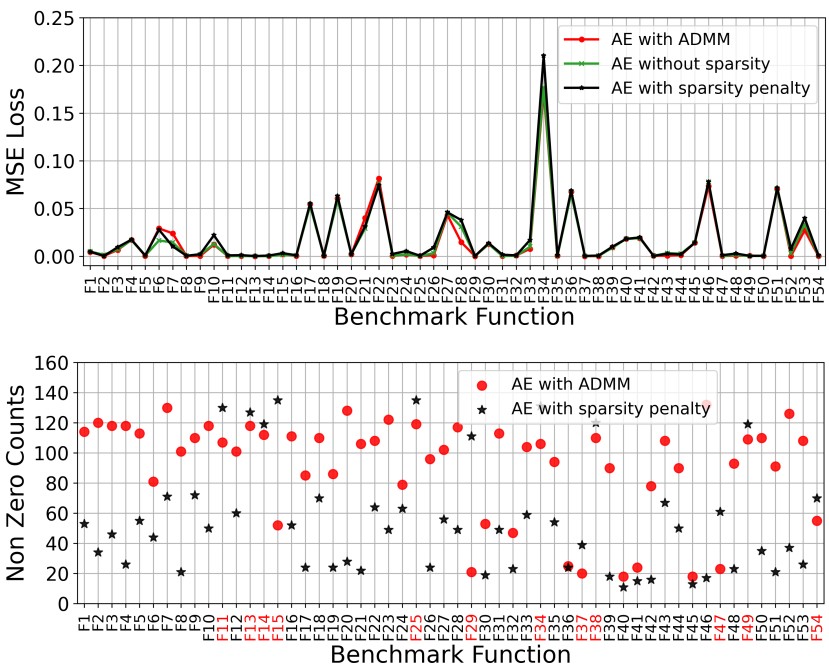

Figure 6: Top: MSE of the best-performing models, comparing AE followed by ADMM, AE followed by sparsity penalty, and AE without pruning. Bottom: Corresponding number of non-zero weights in the selected models. Function labels are color-coded to indicate which method yields a sparse model.

directions define a local 2D subspace around $z_{\text{init}}$, along which we generate perturbed embeddings $z\prime = z_{\text{init}} + \alpha v_1 + \beta v_2$, where $\alpha, \beta \in [-3, 3]$ are sampled on a grid.

Each perturbed embedding $z\prime$, along with $z_{\text{init}}$, is decoded using $D_k$ to obtain the corresponding decoded MLPs. For a fixed set of input values, we compute the outputs of $D_k(z_{\text{init}})$ and $D_k(z\prime)$, treating the former as ground truth. The MSE between the two outputs quantifies the functional deviation induced by perturbations in the embedding space.

We visualize the resulting MSE values over the 2D grid of $(\alpha, \beta)$ to assess the smoothness of the embedding space. Figure 7 shows that the MSE varies smoothly across directions $v_1$ and $v_2$, indicating that the latent space learned by the autoencoder preserves functional continuity.

### F.2 Functional Approximation by the Autoencoder

We visualize the decoded outputs corresponding to input MLPs with 1 to 4 hidden layers, as shown in Figure 8. While the decoded MLPs generally approximate the functional behavior of the inputs, some discrepancies remain, particularly for deeper networks. However, as demonstrated in Section 4 of the main text, our method is still able to discover high-performing MLPs through gradient-based optimization in the embedding space.

We hypothesize that this is because the embedding space only needs to provide a sufficiently smooth and semantically organized landscape, such that gradient descent can navigate toward better-performing regions. The autoencoder does not need to achieve perfect reconstruction, but rather serve as a functional embedding mechanism. Improving the fidelity of the autoencoder would further enhance the smoothness and precision of the search space, leading to improved performance in the downstream optimization.

### F.3 Different hyperparameter choices

In Equation 6, Equation 7, and Equation 10, the sparsity and compactness behavior of the searched neural networks is primarily governed by three key hyperparameters: $\lambda_s$ for controlling the overall

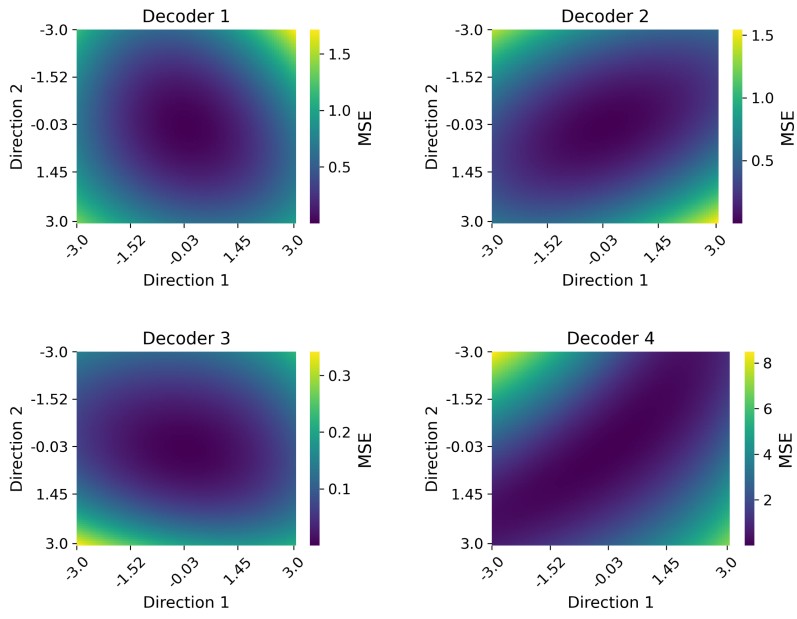

Figure 7: Visualization of latent space smoothness across 2D subspaces for the four decoders.

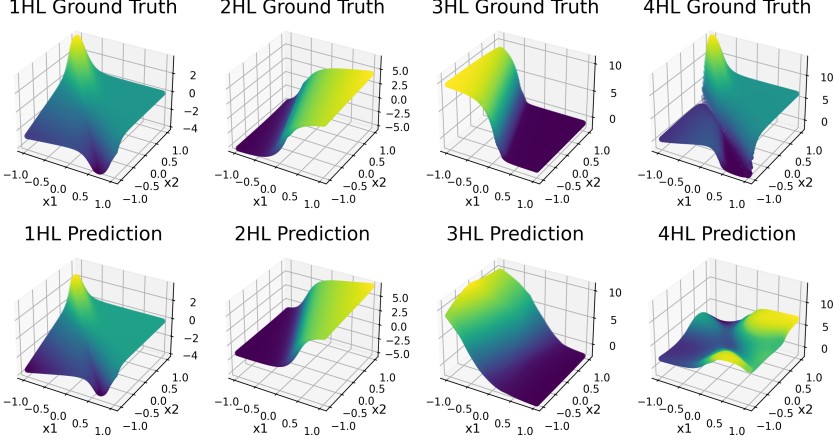

Figure 8: Comparison between the outputs of input MLPs and their best decoded counterparts (minimum-MSE selection among all decoders). HL refers to hidden layers.

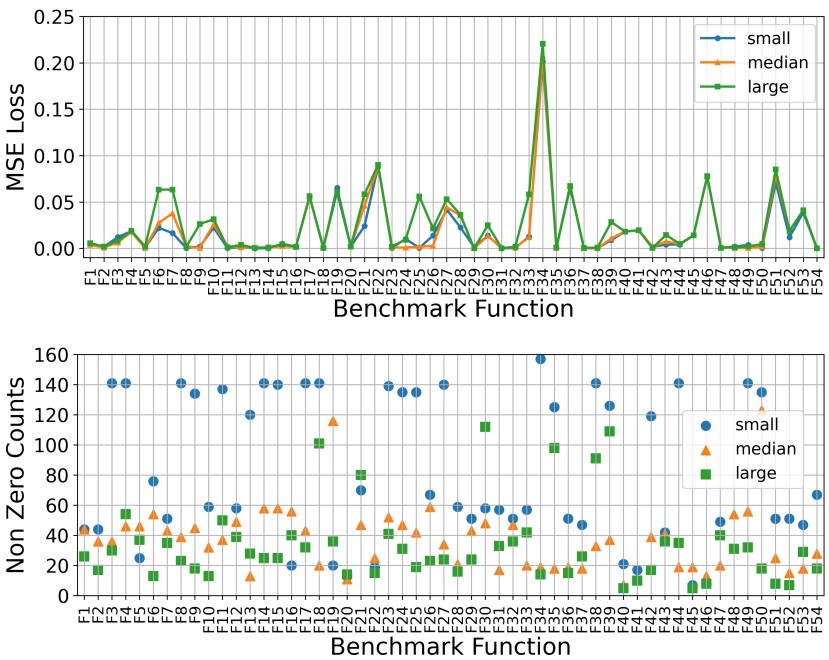

Figure 9: Test MSE (top) and number of non-zero weights (bottom) for the best-performing networks under three different levels of sparsity and compactness regularization.

strength of sparsity, and the internal weighting coefficients $\alpha$ and $\beta$ used in the neuron compactness term. We evaluate three settings with increasing regularization strength, as summarized in Table 1.

Table 1: Hyperparameter settings for different levels of sparsity and compactness regularization.

| Penalty Level | $\lambda_s$ | $\alpha$ | $\beta$ |
|---|---|---|---|
| Small Penalty | $1 \times 10^{-5}$ | $1 \times 10^{-1}$ | $1 \times 10^{-4}$ |
| Medium Penalty | $1 \times 10^{-4}$ | $4 \times 10^{-1}$ | $1 \times 10^{-3}$ |
| Large Penalty | $1 \times 10^{-3}$ | $4 \times 10^{-1}$ | $1 \times 10^{-1}$ |

Figure 9 illustrates the effect of different regularization settings on the performance and sparsity of the resulting models. As expected, the curves align well with the penalty levels defined in Table 1: stronger regularization (i.e., larger values of $\lambda_s$, $\alpha$, and $\beta$) generally leads to lower non-zero weight counts but higher MSE. This confirms the trade-off between model compactness and predictive accuracy. In practice, medium penalty settings tend to offer a favorable balance, yielding compact architectures with minimal loss in performance.

### F.4 Change in size of the embedding space

To assess whether a large embedding dimensionality is necessary, we train autoencoders to embed 1–4 hidden layer MLPs with fixed structure: no bias terms, no activation functions, and exactly 5 neurons per layer. Each autoencoder is trained for 50 epochs, with 640,000 sampled MLPs per epoch. Figure 10 compares performance on all regression tasks across different embedding sizes. In nearly all tasks, the $5 \times 768$ embedding yields lower or equal MSE loss compared to smaller alternatives ($5 \times 256$ and $1 \times 512$). This suggests that a sufficiently large embedding space is beneficial for capturing functional variations in MLPs. Accordingly, we use a $7 \times 768$ embedding in our main experiments.

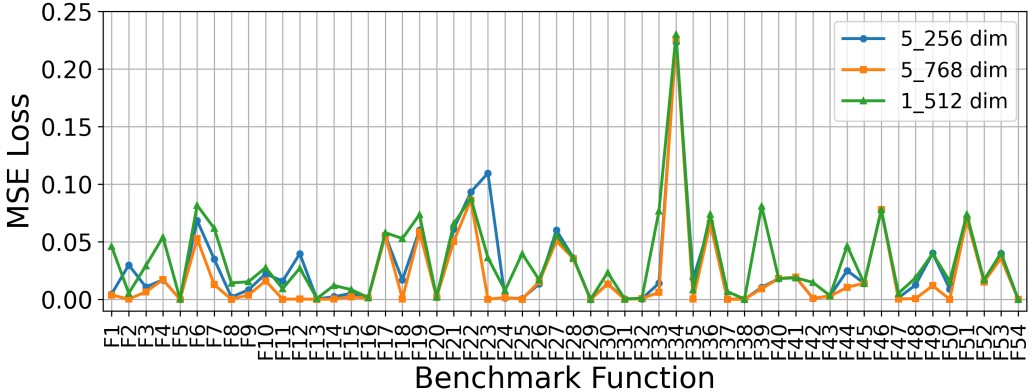

Figure 10: Test MSE for the best MLPs found using embedding vectors of different dimensions on the CORNN dataset. Each curve corresponds to a specific embedding size: $5 \times 768$, $5 \times 256$, and $1 \times 512$.

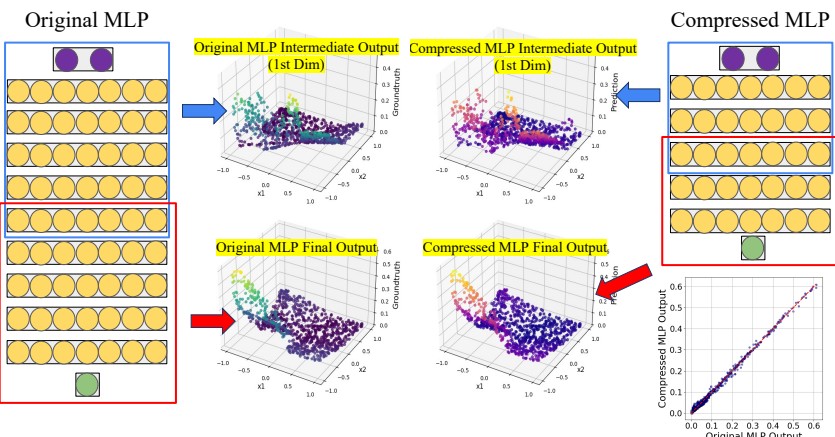

Figure 11: Intermediate and final outputs of the original vs. compressed MLP. The compressed 9 hidden layer MLP closely replicates both internal representations and final predictions of the original 9 hidden layer MLP. This approach also demonstrates SWAT-NN's ability to train variable input/output MLPs.

### F.5    Compression of networks and variable input/output size

It is possible to compress large MLPs into smaller ones using the existing embedding space, without requiring deeper autoencoders. The key idea is that a deep network can be decomposed into smaller sub-networks connected in sequence, each of which can be individually compressed using the autoencoder. As a proof of concept, we compress a 9-hidden-layer MLP into a 5-hidden-layer MLP by splitting it into two sub-networks and compressing each with a lightweight 4-to-2 autoencoder. This autoencoder is trained on MLPs with leaky ReLU activations, 7 neurons per layer, and no biases.

Importantly, our matrix-based representation supports varying input and output sizes by updating the second-channel mask (see Figure 2). This flexibility allows us to compress the first 4-layer MLP with input size 2 and output size 7, and the second with input size 7 and output size 1.

The compression is achieved by optimizing in the latent space of the 4-to-2 autoencoder for each sub-network. As shown in Figure 11, the output of the compressed 5-layer MLP closely matches that of the original 9-layer MLP, demonstrating that large networks can be effectively approximated using SWAT-NN without needing to retrain a deeper autoencoder that matches the size of the original large network.

# References

Hanxiao Liu, Karen Simonyan, and Yiming Yang. DARTS: Differentiable architecture search. In *International Conference on Learning Representations (ICLR)*, 2018. URL `https://openreview.net/forum?id=S1eYHoC5FX`.