# OpenReview forum: "SWAT-NN: Simultaneous Weights and Architecture Training for Neural Networks in a Latent Space"
_NeurIPS.cc/2025/Workshop/UniReps — UniReps2025_

### Official Review · Reviewer_yNMa · 2025-09-07
**A Novel framework for Neural Architecture Search**

**Confidence:** 3

**Review:**

Pro:
The paper proposes a novel framework called SWAT-NN for neural architecture search. It embeds entire neural networks, including both architecture and weights into a continuous latent space using a multi-scale autoencoder. Optimization is then performed via gradient descent directly in this latent space to find an optimal network for a given task, simultaneously tuning its structure, sparsity, and weights.

The method allows for optimization at a very fine-grained level (neuron-level activation functions, individual weight sparsity, neuron pruning) within a single, unified framework. This is more flexible than methods that only search for macro-architectures or layer types.

The incorporation of sparsity and compactness penalties directly into the loss function is effective. The results clearly show that SWAT-NN consistently discovers models with significantly fewer non-zero weights and active neurons than the baselines, which is crucial for efficiency and interpretability.

The method is evaluated on 54 regression tasks, where it demonstrates a strong ability to find compact and sparse models with competitive performance.


The appendix is detailed and well-structured, providing necessary implementation details on the matrix representation, autoencoder training, regularization terms, and additional experiments that bolster the main claims.

Cons:
The major drawback is the need to pre-train a large, multi-scale autoencoder on a massive dataset of randomly generated MLPs. This is a computationally expensive and time-consuming process.

The choice of a DARTS+ADMM baseline for MLPs is reasonable but not perfect. DARTS was designed for cell-based search in CNNs and can be unstable when applied directly to MLPs. A stronger comparison might include other recent weight-sharing NAS methods or Bayesian optimization techniques tailored for small networks.

**Score:**

4

**Topic Fit:**

2

---

### Official Review · Reviewer_ifh8 · 2025-09-15
**Nice idea but issues with evaluations and comparisons.**

**Confidence:** 4

**Review:**

Summary:
The paper proposes SWAT-NN, a two-stage approach. The first stage consists of pretraining a multi-scale autoencoder (encoder + 1–4 GPT-2–style decoders) that embeds full MLPs (structure + weights + per-neuron activations) into a continuous latent space. The second step consists of optimizing, for a new dataset, a latent vector z by gradient descent so that a decoder emits an MLP whose predictions minimize task loss with sparsity/compactness regularizers and temperature-annealed activation selection. Experiments on the CORNN suite (54 synthetic regression tasks) suggest that SWAT-NN reaches similar MSE but with fewer non-zero weights than an adapted DARTS + ADMM baseline.

Strengths

1.	The search conducted in function space, rather than over discrete graphs, with a single latent vector driving both topology and weights, makes the approach clear and straightforward.

2.	The paper presents explicit mechanisms for weight sparsity and neuron pruning, plus per-neuron activation choice with temperature annealing (Section 4.1, Appendix D.1–D.2).

3.	The multi-scale decoder design, consisting of one encoder and several decoders tied to depth, offers a neat way to amortize representation across models of different depths.


Weaknesses

1.	The only baseline that is considered is an adapted DARTS followed by ADMM. There is no comparison to mainstream sparse-training or L0/Hard-Concrete style methods for MLPs, and the two-step model selection can favor the proposed method without reporting full Pareto fronts or aggregate statistics across all tasks.

2.	The universality/generalization of the approach given tight pretraining constraints is questionable. The autoencoder is trained on 2-D inputs, 1-D outputs, and up to 7 neurons per hidden layer with narrow weight/bias ranges. Since only a limited variable-I/O demonstration appears, the claims of dataset-agnostic universality feel premature.

3.	The compute accounting and efficiency are mostly unclear. One epoch of autoencoder training takes approx. 3 hours on a single A5000 with 50k batches, while downstream search is “approx. 1 minute per task per MLP configuration,” but total pretraining epochs/GPU-hours and amortized cost versus the baseline are not reported.

4.	The evaluation scope is limited to synthetic regression. All core results are on CORNN’s synthetic functions. There is no real-world tabular/time-series study or external domain to demonstrate transfer or robustness, which weakens the generalization claim.


Further Comments

1.	The Related Work in Section 2 defines continuous NAS as “constrained by task-specific predictors or datasets,” but NAO and follow-ups already learn continuous encodings. Please position more carefully how the function-level encoding (weights + activations + masks) in this paper differs from architecture encoders and why that matters empirically.

2.	The central claim of simultaneous optimization of architecture and weights is conceptually plausible but not demonstrated empirically. It is recommended to show direct evidence that both evolve during latent optimization (e.g., trajectories of active neuron masks, layer counts, and activation distributions across epochs of optimizing z in Section 3.2).

3.	Per-neuron activation mixtures during training change the function class relative to one-hot activations and may cause train/test mismatch when discretizing at inference. It would be interesting to quantify the accuracy drop when “snapping” the activations, and include annealing schedule ablations (T_init, T_final, E_anneal) on multiple CORNN tasks.

4.	The autoencoder training distribution (weights/bias ranges, input sampling) may not match the downstream data distributions in CORNN. It is recommended to analyze the sensitivity to this mismatch and consider domain randomization (input ranges, noise, weight priors) to mitigate potential overfitting of the representation to synthetic pretraining.

5.	Please clarify whether the CORNN tasks used for evaluation are split into train/val/test and how early stopping/hyperparameter selection interacts with those splits. The current description is unclear and leaves ambiguity that affects the credibility of the comparisons.

6.	Some typos should be corrected. Duplicated phrase “and leaky ReLU, and leaky ReLU” in Section 1. “1 hidden layers” in Fig. 2. Thorough proofreading is required.

**Score:**

2

**Topic Fit:**

2

---

### Official Review · Reviewer_8hx3 · 2025-09-19
**Review of SWAT-NN: Simultaneous Weights and Architecture Training for Neural Networks in a Latent Space**

**Confidence:** 5

**Review:**

### Summary

This paper introduces SWAT-NN, a framework that simultaneously optimizes both neural network architectures and weights. It does so by embedding networks into a continuous latent space using a multi-scale autoencoder. Networks with similar behavior are trained to be close in this space, and gradient descent is then used to find sparse, compact MLPs that work well for regression tasks.

### Strengths

- The idea of using an autoencoder to create a continuous embedding space for simultaneous architecture and weight optimization is creative and represents a departure from traditional NAS approaches.

- The multi-scale decoder design, enabling variable network depths and the incorporation of sparsity/compactness penalties are well-motivated technical choice.

- On the CORNN benchmark, SWAT-NN consistently produces sparser networks than the DARTS+ADMM baseline while maintaining comparable accuracy.

### Weaknesses

- Evaluation is restricted to MLPs on regression tasks only. For broader impact, the method needs validation on classification tasks and more complex architectures (CNNs, Transformers).

- DARTS+ADMM is not a standard approach for this problem. More appropriate baselines would include direct sparse MLP training with regularization or other multi-objective optimization methods.

- The autoencoder approach adds significant complexity compared to simpler alternatives. The paper doesn't convincingly demonstrate when/why this complexity is necessary.

- I also noticed that the paper lacks several key analyses:
  - No computational cost comparisons with simpler methods
  - Limited ablation studies on key design choices
  - No statistical significance testing of results
  - Insufficient analysis of embedding space properties beyond smoothness visualization

- Terms like "fundamentally different" (line 3) and "universal embedding space" (line 46) oversell the contribution of the paper. The universality is only demonstrated within the narrow MLP regression domain.

### Major Concerns

- The functional similarity assumption underlying the autoencoder training (Equation 6, Appendix C line 248) may not hold broadly across different tasks and domains.
- The matrix representation scheme, while comprehensive, seems unnecessarily complex for the relatively simple MLP search space.
- The paper lacks a discussion of failure modes or limitations of the approach.

### Minor Issues

- Key technical details are relegated to appendices, making the main paper harder to follow
- Some figures (especially Figure 3) could be clearer
- Writing could be more precise about the method's limitations and scope

### Suggestions

I would like to suggest a few points for consideration in the next version of the paper:

1. Evaluate classification tasks and CNN/Transformer architectures, or at least include hints on how the method could generalize to other layer types
2. Include comparisons with simpler sparse training baselines
3. Provide a theoretical or empirical analysis of when the autoencoder complexity is justified
4. Add computational cost analysis and statistical significance testing
5. Consider reframing claims about universality to be more precise about the demonstrated scope

**Score:**

3

**Topic Fit:**

2

---

### Official Review · Reviewer_Cmnc · 2025-09-19
**Review of SWAT-NN: Simultaneous Weight and Architecture Training for Neural Networks**

**Confidence:** 3

**Review:**

The paper introduces a novel method for neural architecture search (NAS) that simultaneously optimizes both network weights and architecture. The approach embeds a full MLP into a matrix with an associated mask and trains a GPT-2 autoencoder to encode the model such that the decoded model exhibits similar behavior on inputs. Through this setup, the training process jointly learns both the architecture and the weights. Moreover, the framework supports compactness and sparsity penalties, encouraging the learning of efficient weight representations.

The authors compare SWAT-NN against a DARTS+ADMM baseline that searches for the number of neurons and prunes them, evaluated across 54 regression datasets. The main strength of their results lies in the higher sparsity of networks produced by SWAT-NN.

This work presents a fresh perspective on NAS, and with further development, it could grow into a strong approach. The perspective itself is interesting enough to merit acceptance at a workshop. However, I have several major concerns:

- I am not convinced that the framework can be generalized to more complex neural networks, as the authors suggest in their conclusion. For datasets significantly more complex than simple low-dimensional cases, jointly optimizing both architecture and weights may pose substantial challenges. Additional experiments are necessary to validate the method’s potential in such settings.
- Although the authors report the method’s time complexity, they do not provide a comparison with alternative approaches that optimize only weights or only architectures. It is important to demonstrate that SWAT-NN is competitive in efficiency relative to these other methods.

To conclude, much more extensive evaluation is needed to ensure that SWAT-NN is a practical approach suitable for real-world applications.

### Minor Adjustments
- In line 26, insert a space between "space" and "Santra".
- The phrase ", and leaky ReLU" is repeated in line 39 and should be corrected.

**Score:**

4

**Topic Fit:**

3

---

### Official Review · Reviewer_q66L · 2025-09-19
**Comprehensive work on optimizing neural nets, thorough results on relevant benchmarks, good advances**

**Confidence:** 3

**Review:**

**Summary**:

This paper presents a method for optimizing both the architecture of a neural net and its weights at once, in a shared latent space using autoencoders. I think that the work is obviously very comprehensive, and shows some interesting results. They test their method on a benchmark of different regression tasks, and show that they can add sparsity constraints to produce more efficient models when decoding from their latent space. The paper also provides numerous appendices that go into detail on different aspects of the method.

In my opinion, the paper is obviously worthwhile for an extended abstract, and so I recommend acceptance to the workshop. That being said, I do have some complaints. The primary one, which is almost a double-edged sword, is that there are so many results in the appendices that I think this really should have been an 8-page proceedings track paper instead of an extended abstract paper. I suspect that this is being submitted to ICLR as well, which is fine. But I did feel that I had to read through the appendices to really understand the paper. For what it's worth, I did read through all of them and also looked through the code that was uploaded as well (thank you for that effort). The writing in the appendices was very clear and helped me understand what was going on, but the main body of the work was cramped. Otherwise I have some minor issues, but again nothing to jeopardize acceptance to the workshop. I'll list them out further down. Overall the authors provide a comprehensive method and back it up with loads of experiments. Good work.

**Strengths and Weaknesses**:

1. Quality
	1. 4/4. Very comprehensive, high quality for an extended abstract.
2. Clarity
	1. 3/4. The main body was very condensed, I had to read through the rest of the appendices. That being said, the writing wasn't bad it was just a great deal of content in four pages.
3. Significance
	1. 4/4. A new method for joint optimization of architecture and weights seems highly valuable.
4. Originality
	1. 3/4. I am less familiar with this area, but it does seem to straightforwardly follow from other works, although it is still a novel advance.

**Questions & Limitations**:
Note that I am combining the questions and limitations section of the NeurIPS reviewer guideline into one, because there is no rebuttal period for the workshop.

Here I'll list out some major and minor criticisms. Hopefully they will reach you in time to help prepare you for your next submission. I'm recommending acceptance so none of these are dealbreakers for me, either alone or in total. Good job on your work.

Minor:
- How do you test for generalization on each task? I understand that this is a standard benchmark that may not include this, but if there is no validation on each task that would be helpful for reviewers to know. This may be obvious to someone more familiar with the area however.
- In Fig. 1, the caption says that the color coding highlights the more compact architecture, but it seems that the baseline of DARTS gets more compact than your SWAT-NN method sometimes, and the colors just indicate which architecture is which. So this statement in the caption seems misplaced.
- In section 4.1, you state 'SWAT-NN always identifies solutions closer to the optimal accuracy-sparsity tradeoff'. But out of the 16 randomly sampled functions you show on the left of figure 1, this doesn't always seem to be the case and there are some where the DARTS baseline is closer to the bottom left, or at least equally as close as SWAT-NN. Such as the 'sphere', 'bartles conn', 'sum squares' etc tasks. If you're going to make a statement that X is ALWAYS better than Y, you need to provide some quantification of it. Even if X is better than Y in 90% of cases, I want to see a breakdown of cases where DART is not, at least showing how many of the 54 benchmark functions SWAT is doing better than DART in and by how much.
	- And then in the paragraph immediately below, you mention that 'in most cases, SWAT-NN consistently discovers models with significantly fewer non-zero weights'. So I don't understand why the framing is so different between these two paragraphs, and it makes it seems that two different people wrote each one.

Major:
- You're concentrating on regression models with MLP layers. And the MLP layers themselves are quite small in size, with few neurons each. You have one sentence that calls this out in the conclusion, but I'd prefer to see something which analyzes this slightly more. For now these are all toy tasks, which is fine if that's where the field is at. And this paper is still making worthwhile advances. But connecting your work to something that might touch on architectural optimization for larger, SOTA models would be a good idea.
- The paper is very comprehensive. In my opinion, this should have been an 8-page proceedings track paper, but again I understand if you're deciding to submit to another conference and wanted to go for the non-archival version. For what it's worth, I actually like the appendices and the figures and writing there is really well done. I think it's unfortunate that the main body feels very cramped and I had to read through further to understand everything. In the future, I would recommend taking the related works section and putting that in the appendix, and spending more time on the methods in the main body. Almost half the paper is gone with abstract / intro / related works before we even get to read about the method, which is of course the interesting part.

Again none of these jeopardize the paper. I found it to be interesting and well researched, and recommend acceptance.

**Overall Score**:

4/5, accept.

**Confidence**:

3/5.

**Score:**

4

**Topic Fit:**

2

---

### Official Review · Reviewer_GUb2 · 2025-09-19
**The paper proposes a clear and novel approach that makes progress on a challenging problem. While the evaluation could be broadened and strengthened, the contribution is meaningful and has potential to inspire further research**

**Confidence:** 3

**Review:**

**Summary:**
The paper introduces an approach to jointly optimize neural network weights and architectures by embedding both into a latent space via a multi-scale autoencoder. Optimization in this space enables simultaneous adjustment of structure and parameters. Results on regression benchmarks indicate that the method is able to discover more compact architectures while maintaining accuracy, with modest performance improvements over baselines.

**Pros**
- The method consistently produces smaller models without sacrificing accuracy.
- Represents a promising step toward unified architecture and weight optimization.
- The core idea is clearly articulated and easy to follow.

**Cons**
- The evaluation focuses only on regression tasks; additional justification for not including classification or other tasks would be valuable.
- The results in Fig. 1 (left) could benefit from a deeper explanation of the trade-offs and their practical implications.
- The choice of DARTS+ADMM as the primary baseline is not fully discussed; considering stronger or more diverse baselines could make the evaluation more convincing.

**Corrections**
- Line 39: “and leaky ReLU” is repeated.
- The appendix should include a heading.

**Score:**

4

**Topic Fit:**

3